# A genomic variant of *ALPK2* is associated with increased liver fibrosis risk in HIV/HCV coinfected women

Alec T. McIntosh[1], Renhuizi Wei[1], Jaeil Ahn[2], Brad E. Aouizerat[3], Seble G. Kassaye[4], Michael H. Augenbraun[5], Jennifer C. Price[6], Audrey L. French[7], Stephen J. Gange[8], Kathryn M. Anastos[9], Radoslav Goldman[1,10,11] *

1 Department of Oncology, Georgetown University, Washington, DC, United States of America,
2 Department of Biostatistics, Bioinformatics & Biomathematics, Georgetown University Medical Center, Washington, DC, United States of America, 3 Bluestone Center for Clinical Research, College of Dentistry, New York University, New York, New York, United States of America, 4 Department of Infectious Diseases, Georgetown University Medical Center, Washington, DC, United States of America, 5 Division of Infectious Diseases, Department of Medicine, State University of New York, Downstate Medical Center, Brooklyn, New York, United States of America, 6 Division of Liver Diseases, Department of Medicine, University of California San Francisco, San Francisco, California, United States of America, 7 Division of Infectious Disease, Department of Internal Medicine, Stroger Hospital of Cook County, Chicago, Illinois, United States of America, 8 Department of Epidemiology, Johns Hopkins Bloomberg School of Public Health, Johns Hopkins University, Baltimore, Maryland, United States of America, 9 Departments of Medicine and Epidemiology & Population Health, Albert Einstein College of Medicine, Bronx, New York, United States of America, 10 Department of Biochemistry and Molecular & Cell Biology, Georgetown University Medical Center, Washington, DC, United States of America, 11 Clinical Translational Glycoscience Research Center, Georgetown University Medical Center, Washington, DC, United States of America

* rg26@georgetown.edu

**Data Availability Statement:** As described in the manuscript, the cohort is in active follow-up. The cohort has been identified as one with multiple vulnerabilities (e.g., racial/ethnic minority women,

## Abstract

HIV coinfection is associated with more rapid liver fibrosis progression in hepatitis C (HCV) infection. Recently, much work has been done to improve outcomes of liver disease and to identify targets for pharmacological intervention in coinfected patients. In this study, we analyzed clinical data of 1,858 participants from the Women's Interagency HIV Study (WIHS) to characterize risk factors associated with changes in the APRI and FIB-4 surrogate measurements for advanced fibrosis. We assessed 887 non-synonymous single nucleotide variants (nsSNV) in a subset of 661 coinfected participants for genetic associations with changes in liver fibrosis risk. The variants utilized produced amino acid substitutions that either altered an N-linked glycosylation (NxS/T) sequon or mapped to a gene related to glycosylation processes. Seven variants were associated with an increased likelihood of liver fibrosis. The most common variant, *ALPK2* rs3809973, was associated with liver fibrosis in HIV/HCV coinfected patients; individuals homozygous for the rare C allele displayed elevated APRI (0.61, 95% CI, 0.334 to 0.875) and FIB-4 (0.74, 95% CI, 0.336 to 1.144) relative to those coinfected women without the variant. Although warranting replication, *ALPK2* rs3809973 may show utility to detect individuals at increased risk for liver disease progression.

coinfected). Whereas participants from the cohort who contributed to the findings summarized in this manuscript provided written consent for genetic studies, said consent was collected prior to the most recent guidelines and requirements for data sharing. The WIHS cohort operates under an alternative data sharing plan registered with the National Institutes of Health and access to phenotypic and genomic data can be requested by submitting a Concept Sheet, which can be found along with instructions for Concept Sheet submission, at https://statepi.jhsph.edu/mwccs/. The accession number for the WIHS in dbGaP genomic data is now provided in the manuscript (phs001503). The cohort is currently being re-approached to obtain informed consent for sharing of their data. This has been consistent with other genomic studies in the WIHS cohort.

**Funding:** Research reported in this publication was supported in part by the National Institutes of Health to RG (awards R01CA135069 and R01CA238455). The content is solely the responsibility of the authors and does not necessarily represent the official views of the National Institutes of Health. The MWCCS is funded primarily by the National Heart, Lung, and Blood Institute (NHLBI), with additional co-funding from the Eunice Kennedy Shriver National Institute Of Child Health & Human Development (NICHD), National Human Genome Research Institute (NHGRI), National Institute On Aging (NIA), National Institute Of Dental & Craniofacial Research (NIDCR), National Institute Of Allergy And Infectious Diseases (NIAID), National Institute Of Neurological Disorders And Stroke (NINDS), National Institute Of Mental Health (NIMH), National Institute On Drug Abuse (NIDA), National Institute of Nursing Research (NINR), National Cancer Institute (NCI), National Institute on Alcohol Abuse and Alcoholism (NIAAA), National Institute on Deafness and Other Communication Disorders (NIDCD), National Institute of Diabetes and Digestive and Kidney Diseases (NIDDK). MWCCS data collection is also supported by UL1-TR000004 (UCSF CTSA), P30-AI-050409 (Atlanta CFAR), P30-AI-050410 (UNC CFAR), and P30- AI-027767 (UAB CFAR). The funders had no role in study design, data collection and analysis, decision to publish, or preparation of the manuscript.

**Competing interests:** The authors of this manuscript have the following competing interests: National Heart, Lung, and Blood Institute (NHLBI; https://www.nhlbi.nih.gov) provided grant support to the following co-authors: U01- HL146242 (BEA, JCP); U01-HL146205 (SGK); U01-HL146202 (MHA); U01-HL146245 (ALF); U01-HL146193

# Introduction

The emergence of highly active antiretroviral therapy has transitioned the once acutely fatal human immunodeficiency virus (HIV) infection to a chronic disease. However, the longer survival of persons living with HIV infection presents a new set of morbidities and increased risk for mortality [1–3]. Due to common modes of transmission, those infected with HIV are at higher risk of contracting hepatitis C virus (HCV). Accelerated progression of liver disease in HIV/HCV coinfected patients compared to those with HCV monoinfection is well documented and liver disease has become a leading cause of non-AIDS related death in coinfected individuals [4,5]. The pathogenesis of liver disease in coinfected individuals is multifactorial [6]. Despite substantial progress in identifying risk factors for liver disease progression in coinfected persons, our understanding of risk factors remains incomplete, with genomic factors among those that remain ill-defined. Even with the emergence of direct-acting HCV antivirals, the ability of these agents to regress fibrosis upon HCV clearance remains unclear [7] and the cost of treatment is often inaccessible to at-risk populations [8]. Taken together, these considerations document that liver fibrosis remains a challenge in coinfected and HCV monoinfected patients. In this study, we examined the impact of non-synonymous single nucleotide variants (nsSNV) affecting N-glycosylation both directly at the NxS/T sequons of proteins and indirectly through the enzymes and lectins of glycosylation-related pathways.

Glycosylation is one of the most common and structurally diverse protein modifications and affects protein synthesis, structure, and function [9]. Glycosylation enzymes or the glycoproteins they produce are involved in immune surveillance and host-pathogen interactions [10,11] as well as in the progression of viral liver disease [12]. To elucidate the impacts of glycosylation on the pathogenesis of fibrosis in HIV/HCV coinfected patients, we focused on nsSNV affecting the NxS/T sequons required for the attachment of N-glycans to proteins and thus the potential to change the number of glycans on the protein surface [13,14]. The Women's Interagency HIV Study (WIHS) is a longitudinal natural history study of HIV infection that features a sufficient number of HIV/HCV coinfected participants and biomarkers of liver disease to evaluate the impact of risk factors for liver disease progression [15], including genetic risk factors [16–18]. Given its longitudinal cohort design, the Fibrosis-4 Index (FIB-4) [19] and the AST to Platelet Ratio Index (APRI) [20] were collected in the WIHS cohort as noninvasive surrogate measures of hepatic fibrosis. These measures were used in lieu of serial tissue biopsy as the means with which to evaluate genetic risk factors in HIV/HCV coinfection [21]. A set of 887 nsSNV were extracted from a genome-wide association study performed in the WIHS cohort. Of these, 278 nsSNV produce an amino acid substitution that alters the potential glycosylation of target proteins by altering the number of N-glycans decorating a protein, which we analyzed in relation to surrogate biomarkers of liver fibrosis. Given that previous work demonstrated that genetic variation in glycosylation-related enzymes and lectins can alter kinetics and binding affinities respectively, the remaining 609 of 887 nsSNV were glycosylation-related proteins evaluated in relation to surrogate biomarkers of liver fibrosis in coinfected participants [22,23].

# Methods

## Ethics statement

The parent WIHS study and this sub-study conformed to the procedures for informed written consent approved by institutional review boards (IRB) at all sponsoring organizations and to human-experimentation guidelines set forth by the United States Department of Health and Human Services, and finally reviewed and approved by the local IRBs in the Baltimore/DC

(SJG); and U01-HL146204 (KMA). This does not alter our adherence to PLOS ONE policies on sharing data and materials. The ethical and legal restrictions to the sharing of the data utilized in this manuscript, as well as the steps necessary to obtain the data are detailed in Data Availability Section.

region. The IRBs at the Chicago site were Cook County CORE Center IRB, Cook County Health & Hospitals System IRB, Northwestern University IRB, Northwestern Memorial Hospital IRB, Rush University IRB, Rush-Presbyterian-St. Lukes IRB, University of Illinois at Chicago IRB. The IRBs at the Los Angeles/Southern California site were University of California at Los Angeles Medical Center IRB, Western IRB, University of Hawaii/Kapi'olani Health Research Institute IRB, University of Southern California IRB. The IRB at the New York Brooklyn site was SUNY Downstate Medical Center Institutional Review Board. The IRBs at the New York Bronx site were Albert Einstein College of Medicine IRB, Beth Israel IRB, Mt. Sinai Medical Center IRB. The IRBs at the San Francisco Bay Area site were Alameda Health System IRB, Sutter Health IRB, University of California, San Francisco IRB, California Committee for the Protection of Human Subjects (CPHS). The IRBs at the Washington District of Columbia site were Georgetown University Medical Center IRB, Howard University IRB, Inova Health Systems IRB, Montgomery County Health Department IRB, Whitman Walker Clinic IRB.

## Study population

The WIHS is an active, multicenter prospective study of the natural history of HIV infection among women with or at risk for HIV-infection in the United States. Established in 1994, a total of 4,982 women (3,677 HIV-seropositive) have been enrolled. At semi-annual visits, participants completed socio-demographic and medical questionnaires, laboratory testing, and a limited physical examination. Data included in this analysis include age, race and ethnicity, continuous clinical measures of liver fibrosis (APRI and FIB-4), plasma HIV RNA viral titer, HCV infection status (at all visits), and HCV viral titer (at baseline only) [24]. HCV status was defined as "positive" when participants had both positive HCV antibody and detectable HCV RNA at their baseline visit. Those that were defined as "negative" had no detectable HCV antibody upon testing. Self-reported race and ethnicity was used to define four groups: "White" (non-Hispanic), "African American" (non-Hispanic), "Hispanic", or "Other". In addition to self-reported race and ethnicity, genomic estimates of ancestry were derived using principle component analysis of ancestry informative genetic markers [25].

## Genotyping

Genotype data in WIHS was generated from genomic DNA from peripheral blood mononuclear cells using the Infinium Omni2.5 BeadChip (Illumina, San Diego, CA, USA) [24]. Of the 10,141 genes (602 glycogene + 9,539 NxS/T-containing genes) known to exist in the human genome, data on 1,029 nsSNVs (698 glycogene + 331 NxS/T) spanning 660 genes (349 glycogene + 311 NxS/T) from 2,120 WIHS participants were available for analysis [13]. Of these 2,120 women, 262 participants were either HCV antibody positive in the absence of detectable HCV RNA (N = 156) or HCV serostatus was never assessed at baseline (N = 106) and therefore excluded (S1 Fig). Of the 331 NxS/T nsSNVs, 44 were mono-allelic (non-polymorphic) and therefore excluded. Among the 287 nsSNVs that remained, missing genotypes across 1,858 patients were imputed with the most common genotype for each nsSNVs (mean participants imputed per nsSNV±SD = 5±6.04). Of the 698 glycogene nsSNVs, 75 were monomorphic and excluded. Among the 623 nsSNVs that remained, missing genotypes across 1,858 patients were imputed with the most common genotype for each nsSNV (mean participants imputed per nsSNV±SD = 5±6.21). A minor allele frequency (MAF) threshold ≥0.001 (≥0.1%) was applied using the MAFs of the coinfected population (N = 661) to further isolate SNVs sufficiently frequent to allow for statistical analysis. Using this method, an additional 9 NxS/T and 14 glycogenes were eliminated from analysis. After applying all exclusion criteria, clinical and

genotype (887 glycogene [609 nsSNV, 278 NxS/T nsSNV] spanning 564 genes) data were available for 1,858 women (S1 Fig). The nsSNV reference sequence identifiers (rsID), major (or common) and minor (or rare) alleles, and MAF across serotypes are provided as supplementary tables for the NxS/T (S1 Table) and glycogene (S2 Table) nsSNVs utilized in our analysis.

## Serologic markers of fibrosis

Fibrosis-4 Index (FIB-4) and the AST to Platelet Ratio Index (APRI) were used as measures of hepatic fibrosis as described in previous literature [21]. For the FIB-4 index, scores <1.45 and >3.25 indicate a high negative and a high positive predictive value for advanced fibrosis respectively [19]. For APRI, scores <0.5 have a high negative predictive value for liver disease while scores >0.7 and >2 indicate a high positive predictive value for moderate and severe hepatic fibrosis respectively [20]. Genetic polymorphisms were independently analyzed against each continuous surrogate index (i.e., APRI, FIB-4) at baseline to identify variants with stronger associations that would manifest across both current clinical diagnostic resources.

## Statistical analysis

For descriptive summaries, continuous variables were summarized using means and standard deviations while categorical variables were summarized using frequency counts and percentages. HIV and HCV RNA status were categorized into 4 serostatus groups: both HIV/HCV noninfected, HIV monoinfected, HCV monoinfected, and HIV/HCV coinfected. Distributions of APRI, FIB-4, and HIV RNA viral load among the four serostatus groups were summarized using descriptive statistics and were compared using chi-square analyses or one-way Analysis of Variance (ANOVA). APRI and FIB-4 scores used for analysis were obtained at baseline visits (pre-2003) at a time prior to the broad use of HCV therapy. For each variant from the 278 NxS/T and 609 glycogene nsSNV that met inclusion criteria, a separate multiple variable linear regression model was constructed for the HIV/HCV coinfected patient population for continuous APRI and FIB-4 outcomes, respectively. Explanatory variables included each variant, age, race and ethnicity, HIV viral load, and HCV viral load where HIV and HCV viral load were normalized using $\log_2(x+1)$ transformation. Model results were near indistinguishable when adjusted for either HCV viral load or HCV status. Genomic estimates of racial and ethnicity were estimated using principle component analysis of 185 ancestry informative markers (SNV) selected to differentiate major racial and ethnic groups in the cohort (i.e., European, African, Hispanic) [25] The first three principle components (PC1, PC2, PC3), which explain >90% variation, were utilized to adjust for genomic estimates of race and ethnicity in the aforementioned linear regression models. HIV and HCV viral load were normalized using $\log_2(x+1)$ transformation. To reassure our findings were not sensitive to confounding factors of liver fibrosis, we performed a sensitivity analysis on the linear regression models with adjustment for additional factors including Hepatitis B (HBV) infection status (categorical variable; positive or negative for chronic infection by serology), participant reported alcohol usage (categorical variable; abstainers [0 drinks/week], mild users [1–7 drinks/week], or heavy users [>7 drinks/week]), and Body Mass Index (BMI, a surrogate marker for Non-Alcoholic Fatty Liver Disease (NAFLD)). The false discovery rate (FDR) was used to account for multiple testing (i.e., 887 linear regression each for FIB-4 and APRI). All reported *p*-values are two-sided, and SNV associations with FDR<0.05 were considered statistically significant. To investigate whether the impact of SNV varies across the four serogroups, we examined SNV*serostatus interactions and computed 95% confidence intervals for each SNV genotype*serostatus configuration. Statistical analyses were performed using SAS *version* 9.4 (SAS Institute, Cary, NC, USA) and R *version* 3.3 (R Core Team, Vienna, Austria).

## Results

Of the 75% of women who were HIV seropositive (1386 of 1858), 52% (n = 725) were HIV monoinfected and 48% (n = 661) were HIV/HCV coinfected (Table 1).

The remaining 25% of women (N = 472) were HCV/HIV noninfected and 22% (N = 104) were HCV monoinfected. The proportion of non-Hispanic African-Americans differed significantly between the four serogroups ($P<0.001$) (Table 1). Post-hoc contrasts indicate that, in comparison to non-Hispanic African Americans, Hispanics were less likely (OR = 0.66 [95% CI, 0.499 to 0.881], $P_{FDR}$ = 0.03) and non-Hispanic Whites were more likely (OR = 1.82 [95% CI, 1.272 to 2.609], $P_{FDR}$ = 0.006) to be in the coinfected serogroup as compared to the HIV monoinfected group. HCV-infected women were older on average ($P<0.001$) than those not infected with the virus (Table 1). Post-hoc contrasts indicate that in comparison to the noninfected serogroup, older subjects were more likely to be monoinfected with HIV (OR = 1.08 [95% CI, 1.059 to 1.100], $P_{FDR}$ <0.001), monoinfected with HCV (OR = 1.21 [95% CI, 1.172, to 1.252], $P_{FDR}$ <0.001) or coinfected (OR = 1.22 [95% CI, 1.197 to 1.252], $P_{FDR}$ <0.001). Distribution of the APRI and FIB-4 fibrotic indexes, stratified by their clinically relevant cutoffs, also differed between the groups ($P<0.001$) (Table 1). Six pairwise comparisons of mean liver scores were performed between designated serogroups for each fibrosis metric (Fig 1). On average, coinfected participants had higher APRI and FIB-4 measures when compared to the HCV monoinfected population (APRI, 0.36, 95% CI, 0.064 to 0.651; FIB-4, 0.78, 95% CI, 0.375 to 1.184) and the HIV monoinfected population (APRI, 0.64, 95% CI, 0.487 to 0.786; FIB-4, 1.19, 95% CI, 0.984 to 1.397) (Fig 1). All other comparisons between serogroups were found to be statistically significant (95% confidence interval) except the mean APRI scores for noninfected vs. HIV monoinfected (-0.15, 95% CI, -0.325 to 0.031) and HIV monoinfected vs. HCV monoinfected (-0.28, 95% CI, -0.571 to 0.012) which follow the expected trend but do not achieve statistical significance (Fig 1). With regard to possible confounding factors, BMI, HBV serological status, and alcohol usage are also shown stratified between the four HIV/HCV serostatuses (Fig 1).

Utilizing the larger cohort for race and age adjustment helped to ensure that we had the population size to account for the impact of these variables on liver fibrosis. Whereas the first principle component of the PCA of genomic markers of ancestry (PC1) appeared to adequately differentiate African-Americans (non-Hispanic), Hispanics, and Whites (non-Hispanic) (Fig 2A), we took a conservative approach and included the first three principle components (PC1, PC2, PC3). Although adjusting for race/ethnicity using self-report identified the same genetic associations, we opted to employ the 3 PCs to better account for confounding that can occur with self-reported race and ethnicity. For example, many participants self-reporting as "Hispanic" or "Other" in the coinfected subgroup present with a genetic background more consistent with the African-American (non-Hispanic) and White (non-Hispanic) clusters (Fig 2B).

After adjustment for multiple testing, seven nsSNV mapping to separate genes met the *a priori* criterion of $P_{FDR}$ <0.05 for at least one of the biomarkers of liver fibrosis (Table 2). Of these, only rs52828316 (*MAN2A2*) was significant by $P_{FDR}$ for both indices. For all nsSNV, two copies of the minor allele (i.e., minor allele homozygotes) was associated with increases in APRI and/or FIB-4 (Table 2). Upon sensitivity analysis of additional adjustable factors of HBV Status, alcohol usage, and BMI (N = 616), all statistically significant nsSNVs were maintained, and APRI and FIB-4 estimates for all but rs1800472 (*TGFB1*) were within 10% of that of the initial model (S3 Table). Two additional nsSNV were found to be significant when additionally adjusting for HBV status, alcohol, and BMI, namely rs3745925 (MADCAM1) and rs2307145 (IL12RB2) (S3 Table). *ALPK2* rs3809973 was evaluated further in relation to hepatic fibrosis

**Table 1. Baseline population characteristics of patients by serostatus.**

| Variable | Noninfected HIV-/HCV- | Monoinfected HIV+/HCV- | Monoinfected HIV-/HCV+ | Coinfected HIV+/HCV+ | |
|---|---|---|---|---|---|
| Levels | N = 368 | N = 725 | N = 104 | N = 661 | *P* |
| *Age at BL, μ ± SD* | 29.6 ± 8.0 | 33.2 ± 7.4 | 39.2 ± 7.2 | 39.8 ± 6.2 | <0.001 |
| *HIV load\*, μ ± SD* | 0 | 11.3 ± 4.4 | 0 | 14.0 ± 3.8 | <0.001 |
| *HCV load\*, μ ± SD* | 0 | 0 | 19.1 ± 3.3 | 20.5 ± 2.8 | <0.001 |
| *Albumin (g/dL), μ ± SD* | 4.2 ± 0.4 | 4.1 ± 0.4 | 4.3 ± 0.4 | 4.0 ± 0.5 | <0.0001 |
| *Race, n(%)* | | | | | |
| Non-Hispanic White | 47 (12.8) | 69 (9.5) | 13 (12.5) | 111 (16.8) | <0.001 |
| Non-Hispanic A-A | 205 (55.7) | 400 (55.2) | 63 (60.6) | 416 (62.9) | |
| Hispanic | 101 (27.5) | 227 (31.3) | 23 (22.1) | 123 (18.6) | |
| Other | 15 (4.1) | 29 (4.0) | 5 (4.8) | 11 (1.7) | |
| *BMI, n(%)* | | | | | |
| <18.5: Underweight | 8 (2.2) | 17 (2.3) | 2 (1.9) | 26 (3.9) | <0.001 |
| 18.5–24.9: Normal | 128 (34.8) | 237 (32.7) | 33 (31.7) | 275 (41.6) | |
| 25–29.9: Overweight | 102 (27.7) | 215 (29.7) | 25 (24.0) | 183 (27.7) | |
| ≥30: Obese I-III | 121 (32.9) | 249 (34.3) | 37 (35.6) | 138 (20.9) | |
| NA | 9 (2.5) | 7 (1.0) | 7 (6.7) | 39 (5.9) | |
| *Alcohol, n(%)* | | | | | |
| Abstainer | 143 (38.9) | 387 (53.4) | 41 (39.4) | 304 (46.0) | <0.001 |
| Mild | 169 (45.9) | 280 (38.6) | 34 (32.7) | 222 (33.6) | |
| Heavy | 53 (14.4) | 43 (5.9) | 24 (23.1) | 121 (18.3) | |
| NA | 3 (0.8) | 15 (2.1) | 5 (4.8) | 14 (2.1) | |
| *APRI, n(%)* | | | | | |
| <0.5 | 367 (99.7) | 633 (87.3) | 64 (61.5) | 267 (40.4) | <0.001 |
| 0.5–0.7 | 0 | 39 (5.4) | 13 (12.5) | 135 (20.4) | |
| >0.7 | 1 (0.3) | 53 (7.3) | 27 (26.0) | 259 (39.2) | |
| *FIB-4, n(%)* | | | | | |
| <1.45 | 366 (99.5) | 657 (90.6) | 79 (76.0) | 336 (50.8) | <0.001 |
| 1.45–3.25 | 2 (0.5) | 59 (8.1) | 20 (19.2) | 255 (38.6) | |
| >3.25 | 0 | 9 (1.2) | 5 (4.8) | 70 (10.6) | |
| *HBV status, n(%)* | | | | | |
| Negative, Non-infected | 248 (67.4) | 457 (63.1) | 39 (37.5) | 147 (22.2) | <0.001 |
| Negative, Anti-HBc Only | 8 (2.2) | 40 (5.5) | 28 (26.9) | 256 (38.7) | |
| Negative, Immunized | 63 (17.1) | 113 (15.6) | 5 (4.8) | 28 (4.2) | |
| Negative, Resolution | 41 (11.1) | 99 (13.7) | 32 (30.8) | 217 (32.8) | |
| Positive, Chronic Infection | 4 (1.1) | 15 (2.1) | 0 | 13 (2.1) | |
| NA | 4 (1.1) | 1 (0.07) | 0 | 0 | |
| *HIV treatment, n(%)* | | | | | |
| No therapy | 0 | 290 (40.0) | 0 | 252 (38.1) | <0.001 |
| Mono therapy | 0 | 47 (6.5) | 0 | 220 (33.3) | |
| Combination therapy | 0 | 88 (12.1) | 0 | 161 (24.4) | |
| HAART | 0 | 300 (41.4) | 0 | 26 (3.9) | |
| NA | 368 (100.0) | 0 | 104 (100.0) | 2 (0.3) | |

Percentages based off columns.

\*Viral loads are $\log_2$ transformed (copies/mL).

**Abbreviations**: BL, Baseline; A-A, African-American; BMI, Body Mass Index; HAART, highly active antiretroviral therapy.

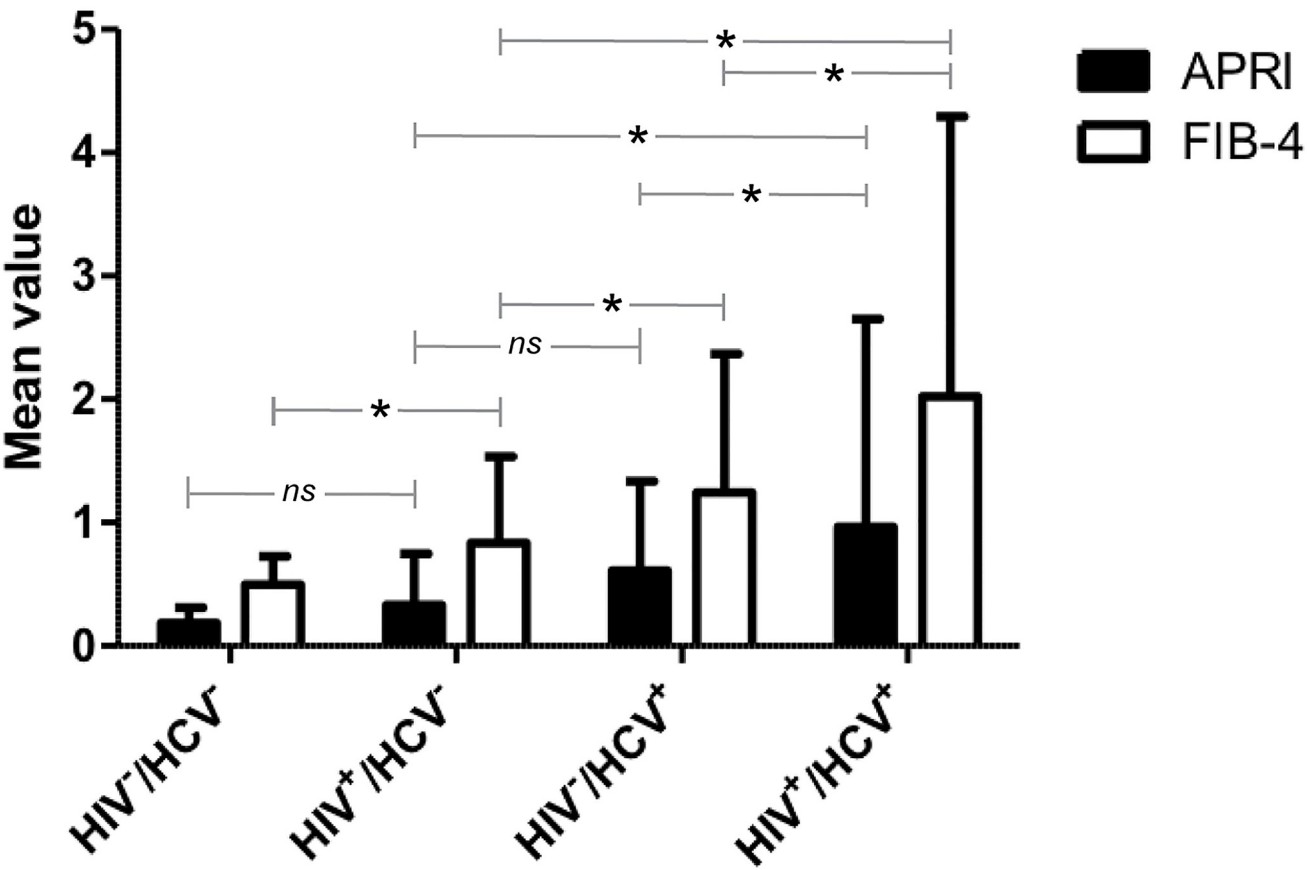

**Fig 1. Mean liver fibrosis scores for each viral serotype.** The mean APRI and FIB-4 scores at baseline visit + SD are shown for 1,858 women stratified by definitive viral serostatus. The number of patients (n) for each viral serostatus are indicated in Table 1. Pairwise comparisons of liver outcomes were performed for all viral-serotype group-comparisons within each liver metric. Apart from those comparisons labeled not significant (*ns*), all other comparisons within fibrosis index aside from those labeled as significant were also below the 0.05 p-value threshold.

among the four viral serogroups because it was the only variant sufficiently frequent (n>100) to result in all three genotypic groups (i.e., major allele homozygotes, heterozygotes, minor allele homozygotes), and the ALPK2 variant was not sensitive to adjustable factors on the linear regression model.

In order to see if the *ALPK2* variant's tentative association with increased liver fibrosis in coinfected women was preferentially impacted by the effects of either virus, we analyzed each genotype of the *ALPK2* variant across all viral serostatuses utilized in the cohort. For the *ALPK2* rs3809973, HIV/HCV coinfected participants who were homozygous for the minor allele had significantly higher mean APRI and FIB-4 scores relative to the coinfected participants homozygous for the major allele (APRI, 0.61, 95% CI, 0.334 to 0.875; FIB-4, 0.74, 95% CI, 0.336 to 1.144) and compared to those HCV monoinfected participants homozygous for the minor allele (APRI, 0.79, 95% CI, 0.370 to 1.200; FIB-4, 1.24, 95% CI, 0.820 to 1.650) (Fig 3A and 3B). Evaluation of the association of ALPK2 rs3809973 with APRI and FIB-4 by racial and ethnic group (i.e., non-Hispanic African-American, non-Hispanic Caucasian, Hispanic) revealed similar patterns of association with one exception in non-Hispanic African-American where no difference by genotypic group was observed with FIB-4 (data not shown). The heterozygotes of coinfected individuals displayed no significant increases in APRI or FIB-

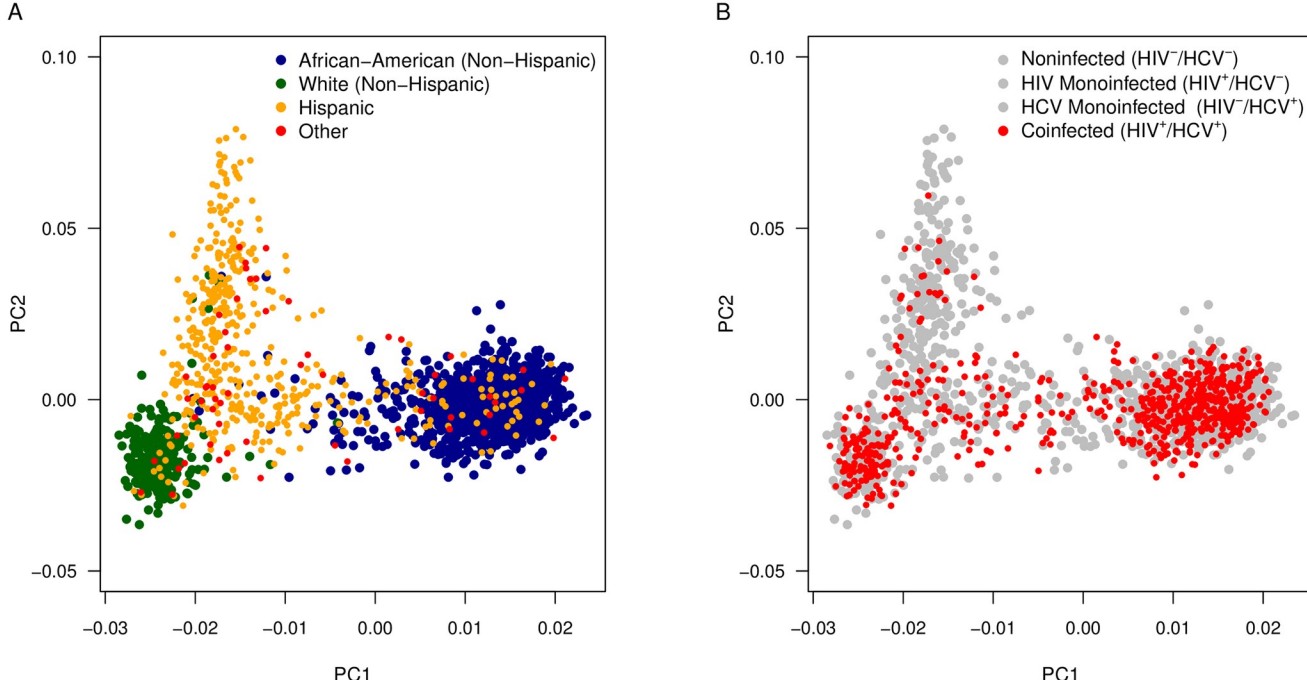

**Fig 2. Genomic estimates of race/ethnicity using principle component analysis.** The 1,858 WIHS participants utilized in this study were stratified using 185 ancestral SNV markers from across the genome. PC1 and PC2 are depicted above. Genomic estimates of race and ethnicity were compared to reported race and ethnicity (A) and the distribution of HIV/HCV coinfected participants of focus are displayed (B).

4 relative to the coinfected major allele homozygotes (APRI, 0.10, 95% CI, -0.128 to 0.332; FIB-4, 0.15, 95% CI, -0.195 to 0.492). These coinfected population findings for the *ALPK2* nsSNV (rs3809973) were observed against a background of similar CD4$^+$ T-cell percentage, measured relative to other leukocyte types, and detectable HIV viral loads at baseline (Fig 3C and 3D). HIV monoinfected participants homozygous for the minor allele displayed significantly increased FIB-4 (0.45, 95% CI, 0.028 to 0.879) scores relative to noninfected participants homozygous for the minor allele, but the finding was not significant for the same comparison using APRI (0.00, 95% CI, -0.422 to 0.429) (Fig 3A and 3B). For the HIV monoinfected genotype, a slight significant difference in the CD4$^+$ T-cell percentage ($P = 0.007$) accompanied this FIB-4 finding. Impacts of variant *ALPK2* allele burden between genotypes in the noninfected and HCV monoinfected serogroups were unremarkable (Fig 3A and 3B). A parallel set of analyses were conducted with adjustment for the additional factors of HBV status, alcohol use, and BMI. Overall, little differences from the present analysis was observed (S2 Fig). However, when comparing the heterozygotes of coinfected individuals to the coinfected major allele homozygotes, FIB-4 was found to be significantly increased (0.19, 95% CI, 0.006 to 0.382) with the APRI measure of the same comparison borderline insignificant (0.17, 95% CI, -0.022 to 0.353) when adjusting for these additional factors (S2 Fig).

## Discussion

This study identified several novel genetic associations among glycogenes and biomarkers of liver fibrosis. Average liver fibrosis scores in the coinfected serogroup analyzed at baseline were significantly higher than that of either HIV or HCV monoinfected participants as shown in Fig 1, recapitulating the fibrotic trends described in the literature for coinfected populations

**Table 2. Baseline screen of multiple linear regression models.**

| Model Fit (N = 661) | | | SNV + Age + PC1-3 + HIV Load + HCV Load[†] | | | | | | Study Sample MAF |
|---|---|---|---|---|---|---|---|---|---|
| Outcome | | | APRI | | | FIB-4 | | | |
| SNV | Alleles (N) | | Estimate | $P$ | $P_{FDR}$ | Estimate | $P$ | $P_{FDR}$ | |
| rs3809973 | CC | (154) | 0.61 | <0.01 | **<0.01** | 0.74 | <0.01 | 0.08 | 0.48 |
| *ALPK2* | AC | (321) | 0.10 | 0.39 | | 0.15 | 0.40 | | |
| (K829N)[‡] | AA | (186) | 0 | | | 0 | | | |
| rs17035120 | TT | (4) | 2.00 | <0.01 | 0.42 | 3.77 | <0.01 | **0.05** | 0.07 |
| *GLT8D2* | TC | (81) | -0.03 | 0.84 | | 0.16 | 0.48 | | |
| (A37T)[‡] | CC | (576) | 0 | | | 0 | | | |
| rs1800450 | AA | (6) | 1.52 | <0.01 | 0.15 | 3.28 | <0.01 | **0.01** | 0.06 |
| *MBL2* | AG | (66) | -0.32 | 0.06 | | -0.42 | 0.10 | | |
| (G54D) | GG | (589) | 0 | | | 0 | | | |
| rs77452813 | TT | (2) | 3.40 | <0.01 | 0.08 | 5.90 | <0.01 | **0.01** | 0.04 |
| *MCOLN2* | TC | (50) | -0.13 | 0.50 | | -0.16 | 0.58 | | |
| (V20I) | CC | (609) | 0 | | | 0 | | | |
| rs52828316 | AA | (2) | 7.55 | <0.01 | **<0.01** | 6.43 | <0.01 | **<0.01** | 0.03 |
| *MAN2A2* | AG | (42) | -0.24 | 0.24 | | -0.30 | 0.33 | | |
| (R1039H) | GG | (617) | 0 | | | 0 | | | |
| rs2228015 | TC | (20) | 1.02 | <0.01 | 0.08 | 1.75 | <0.01 | **0.01** | 0.02 |
| *CCR7* | TT | (641) | 0 | | | 0 | | | |
| (M7V) | | | | | | | | | |
| rs1800472 | TC | (13) | 1.77 | <0.01 | **<0.01** | 1.26 | 0.02 | 0.66 | 0.01 |
| *TGFB1* | CC | (648) | 0 | | | 0 | | | |
| (T263I) | | | | | | | | | |

Bold Indicates Significant *P* Values.

[†] Multiple Linear Regression of each SNV adjusted for Age (continuous), Race (PC1-PC3), HIV/HCV Load (log$_2$ copies/mL; continuous).

Single Letter Amino Acid Abbreviation and Position of Resulting Substitution.

[‡]Potential Gain or Loss of an NxS/T Caused by Substitution.

**Abbreviations**: PC, Principle Component; NxS/T, N-linked Glycosylation Sequon; SNV, Single Nucleotide Variant; MAF, Minor Allele Frequency, FDR, False Discovery Rate.

[6,26]. Of the 887 nsSNV assessed that either directly (nsSNV affecting NxS/T sequons) or indirectly (nsSNV in glycosylation or lectin genes) alter glycosylation pathways or products, we found seven nsSNV that were associated with an increased risk of hepatic fibrosis among the HIV/HCV coinfected population. We observed higher APRI and FIB-4 values (indicative of greater fibrosis) in coinfected participants homozygous for the *ALPK2* rs3809973 minor allele when compared with participants homozygous for the *ALPK2* rs3809973 major allele irrespective of HIV and HCV viral load, BMI, HBV Status, or alcohol usage (Fig 3 and S2 Fig). As elevations in viral HIV titers have been correlated with increased liver injury in the coinfected [27], our findings suggest that the genetic risk of hepatic fibrosis among coinfected individuals carrying the *ALPK2* rs3809973 risk allele is not confounded by these titers. The lack of association in coinfected participants between the *ALPK2* variant heterozygote and increased liver fibrosis suggested the need for variant homozygosity for the detrimental impacts of the variant to manifest in the coinfected and conforms with a recessive mode of inheritance. As shown in Fig 3, among HIV monoinfected participants, the relative CD4$^+$ T-cell percentage among leukocytes was inversely associated with FIB-4. Furthermore, coinfected participants that were homozygous for the minor allele were significantly increased in terms of fibrosis risk

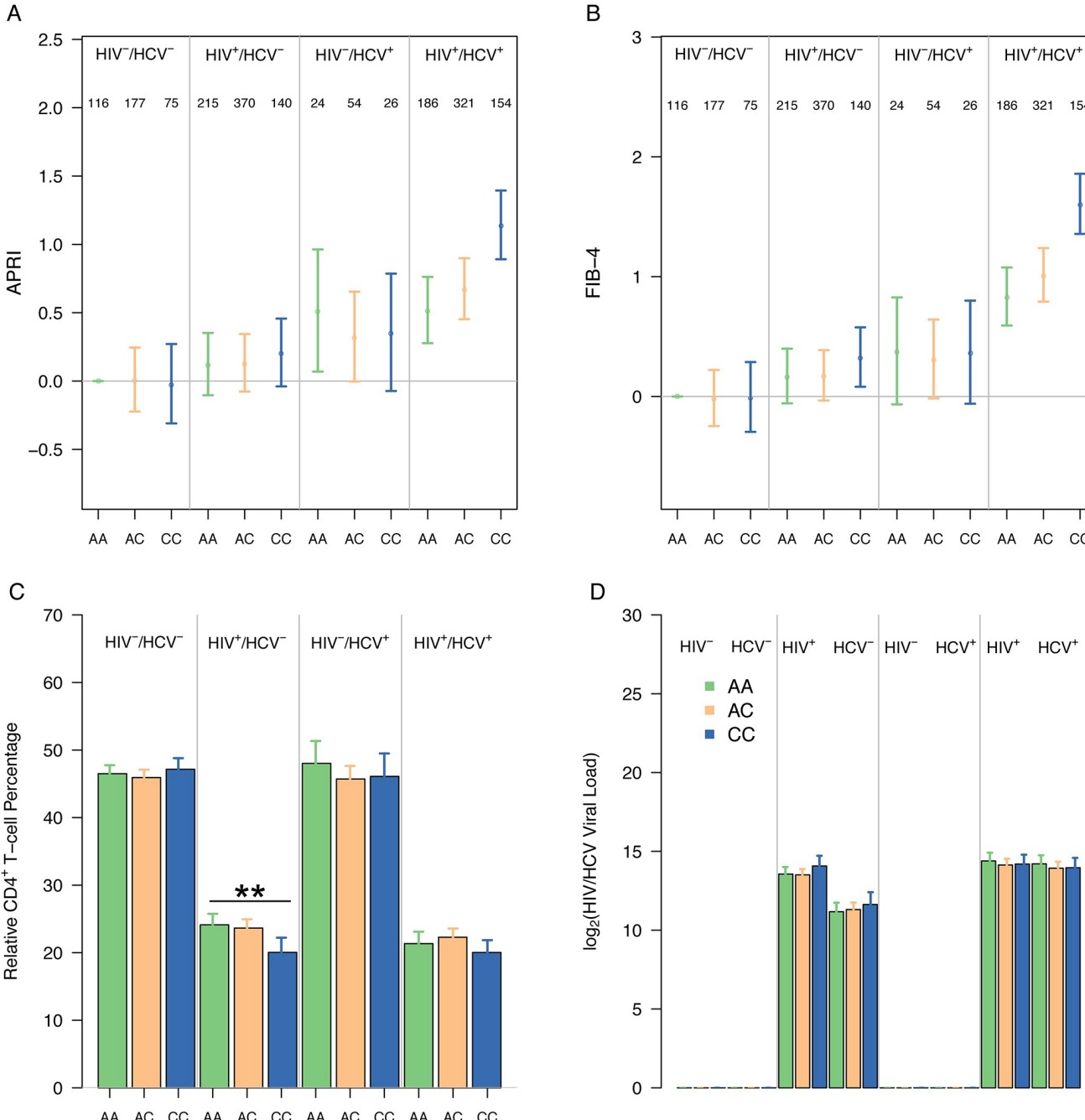

**Fig 3. rs3809973 (ALPK2) impact on APRI and FIB-4 outcomes.** Mean baseline APRI (A) and FIB-4 (B) score with 95% confidence interval shown for the allele pairs of each serotype (Total N = 1,858). The number of samples representing each genotype for the respective serogroups is displayed as follows: homozygous for the major allele (green), heterozygous (orange), and homozygous for the minor allele (blue). Comparisons were shown, for reference, relative to the major allele homozygote of the noninfected serogroup. ANOVA was used to compare relative CD4$^+$ T-cell percentages (C) or HIV/HCV viral loads (D) between the genotypes of each serogroup. Asterisks indicate two-sided P-values below 0.05 (*) and 0.01 (**) respectively.

when compared to HCV monoinfected participants of the same genotype; suggesting a need for general HIV-associated immunosuppression to produce the detrimental fibrosis risk increase. Our comparisons of *ALPK2* rs3809973 among serogroups at baseline therefore leads us to speculate that the risk allele's impact on liver fibrosis is reliant on both HCV-mediated liver damage and its perpetuation by HIV-mediated immunosuppression.

The *ALPK2* gene and rs3809973 have not been linked to a pathological liver phenotype to date. Although the association of *ALPK2* rs3809973 with fibrosis in the setting of HIV/HCV coinfection warrants replication, some circumstantial evidence suggests a potential role for *ALPK2* in risk for viral hepatic fibrosis. Hepatic Stellate cells (HSC) lining the perisinusoidal space of the liver are the fibrogenic cell type responsible for extracellular matrix deposition and subsequent fibrosis in the setting of hepatic inflammation. Although this fibrogenic response is designed to protect against liver injury and typically reverses after the hepatic insult subsides, progressive inflammation and the chronic activation of HSCs can lead to cirrhosis and an increased risk of developing hepatocellular carcinoma [28,29]. Signaling molecules and pathways involved in HSC activation have been described, but most notable is the expression of toll-like receptors on HSCs enables their activation upon exposure to structurally conserved microbial-derived products such as lipopolysaccharide [30]. Although incompletely characterized, the ALPK2 protein was first identified in cardiomyocytes as an integral regulator of heart development [31]. However, most studies of *ALPK2* have focused on the pathological processes of the gastrointestinal tract. *ALPK2* is associated with luminal apoptosis in colorectal cancer cell lines [32]. Luminal shedding is a phenomenon of the enteric innate immune system designed to maintain the function of the gut barrier by preventing the invasion of virulent microbes into systemic circulation. The increased apoptosis of enterocytes in the absence of their concomitant replacement can yield an increasingly permeable intestinal membrane that leaves the host vulnerable to augmented translocation of microbial products [33]. Since increased microbial translocation is a hallmark of HIV infection and lipopolysaccharide activates HSCs [34], any augmentation of this translocation process would exacerbate liver fibrosis in the context of existing hepatitis. In addition, a genome-wide association study of inflammatory bowel disease, a condition in which pathological epithelial shedding is observed, found the mRNA of *ALPK2* to be up-regulated in inflamed mucosa compared to control samples [35]. Therefore, we speculate that *ALPK2* rs3809973 could enhance liver fibrosis by interfering with enteric immunity. Even though this variant generates a novel NxS/T sequon, augmentation of glycan attachment onto the amino acid substitution encoded by the minor allele has not been demonstrated. Structural studies indicate that the variant does not interrupt the kinase domain nor any residues known to be post-translationally modified, glycosylation or otherwise [31,36]. Therefore, the impact of the K829N substitution encoded by *rs3809973* on ALPK2 structure and function warrant investigation.

Although the low minor allele frequency of the remaining 6 nsSNV associated with increased liver fibrosis (i.e., *CCR7* rs2228015, *GLT8D2* rs17035120, *MAN2A2* rs52828316, *MBL2* rs1800450, *MCOLN2* rs77452813, and *TGFB1* rs1800472) precluded detailed modeling, their previous associations with liver disease, metabolism, and viral immunity in other literature provide impetus for further investigation. *TGFB1* has long been linked to enhanced hepatocyte destruction and HSC activation [37]. *GLT8D2* and *MAN2A2* have been implicated in non-alcoholic fatty liver disease [38] and coinfected liver disease [39], respectively. Finally, *CCR7*, *MBL2*, and *MCOLN2* have all been connected to the modulation of host cellular and immune responses in viral infection [40–42]. Taken together, all seven nsSNV and their cognate genes have plausible roles in liver disease pathology and warrant replication in an independent sample(s).

We note that there are limitations to this study. As this is an exclusively female cohort, future studies should aim to validate these findings in men. Although we identified that there are potentially more than 42,000 nsSNV that either directly altered N-linked glycosylation sites (NxS/T) or that may indirectly alter the function of enzymes or lectins interacting with glyco-proteins, only a subset (2.5%) were available for analysis in the cohort due to a combination of low nsSNV allele frequency in the population and/or lack of inclusion of the majority of nsSNV on the commercial array used. A number of factors hindered our attempts at a longitu-dinal assessment of the genetic risk factors on the coinfected population. For one, as the stan-dard of care for HIV treatment changed for coinfected WIHS participants longitudinally (S3A and S3B Fig), adjusting for shifting antiviral treatment regimens at different time points to meet this standard of care was a challenge. Secondly, as this cohort was initially designed to investigate HIV progression, the HCV virologic status was only determined at baseline and HCV clearance was not assessed further, thereby making longitudinal analyses of HCV-infec-tion status susceptible to misclassification bias. In addition, we had no way of determining the duration of HCV infection at the time of serologic assessment. Along similar lines, survival bias and the availability of more effective and less toxic HIV treatments for new enrollees over time in WIHS may have attenuated the relationship between susceptibility alleles and fibrosis [27]. The rate of loss to follow-up was also problematic in that nearly one third of the coin-fected participants at baseline were no longer in the cohort after four visits (2 years) (S3C and S3D Fig). With regard to our sensitivity analysis, only 32 patients (2%) were chronically infected with HBV and this may not have been sufficient power to account for the influence of HBV on liver fibrosis in the linear regression model. Although BMI is a practical means of esti-mating obesity, it was used in the sensitivity analysis as an imperfect measure of NAFLD and may not fully account for the impact of the condition of liver fibrosis. Along similar lines, alco-hol usage was self-reported and may be subject to under reporting in the sensitivity analysis, as the largest fraction of subjects from each of the virally infected serogroups in Table 1 reported to abstain from all alcohol consumption. Finally, APRI and FIB-4 scores are correlated with liver fibrosis risk, but both are imperfect surrogate measures. The utility of these surrogate fibrosis metrics has been validated against biopsy and other measures (i.e. transient elastogra-phy) in previous studies [43], but were not validated by such methods at present.

Given the goal of improving the prognosis of liver disease in the HIV/HCV coinfected pop-ulation, we have identified candidate genes that may participate in hepatic fibrosis. Although these genetic associations require replication, demonstration of the impact of each of the seven nsSNVs on N-linked glycosylation of the site and its cognate protein warrants investigation. Longitudinal HCV testing and assessment of viremia along with additional studies in other HIV/HCV coinfected cohorts may permit better stratification of serogroups and to perform longitudinal analysis of these candidate gene nsSNV.

## Supporting information

**S1 Fig. Genotyping and data processing workflow.**
(TIF)

**S2 Fig. rs3809973 (ALPK2) impact on APRI and FIB-4 outcomes with additional HBV, alcohol, and BMI adjustment.** Mean baseline APRI (A) and FIB-4 (B) score with 95% confi-dence interval shown for the allele pairs of each serotype (Total N = 1,790). The number of samples representing each genotype for the respective serogroups is displayed as follows: homozygous for the major allele (green), heterozygous (orange), and homozygous for the minor allele (blue). Comparisons were shown, for reference, relative to the major allele homo-zygote of the noninfected serogroup. Different from Fig 3, Outcome was additionally adjusted

for factors influencing liver fibrosis including HBV infection status, Alcohol usage, and BMI. ANOVA was used to compare relative CD4$^+$ T-cell percentages (C) or HIV/HCV viral loads (D) between the genotypes of each serogroup. Asterisks indicate two-sided P-values below 0.05 (*) and 0.01 (**) respectively.
(TIF)

**S3 Fig. Longitudinal trends characterizing the HIV/HCV coinfected population.** The values are summarized annually at every second visit. (A) Composition of the anti-HIV drug regimens changing in line with treatment guidelines; no therapy (red), monotherapy (blue), combination therapy (green), or highly active antiretroviral therapy (HAART) (purple). (B) Mean CD4$^+$ percentages (black symbols) and mean HIV viral load (red symbols). (C) The median APRI and (D) FIB-4 indexes.
(TIF)

**S1 Table. NxS/T nsSNV utilized in analysis.**
(XLSX)

**S2 Table. Glycogene nsSNV utilized in analysis.**
(XLSX)

**S3 Table. Baseline screen of multiple linear regression models with additional adjustment for Alcohol, HBV, and BMI (NAFLD).**
(DOCX)

## Acknowledgments

Data in this manuscript were collected by the Women's Interagency HIV Study, now the MACS/WIHS Combined Cohort Study (MWCCS). The contents of this publication are solely the responsibility of the authors and do not represent the official views of the National Institutes of Health (NIH). MWCCS (Principal Investigators): Atlanta CRS (Ighovwerha Ofotokun, Anandi Sheth, and Gina Wingood), U01-HL146241; Bronx CRS (Kathryn Anastos and Anjali Sharma), U01-HL146204; Brooklyn CRS (Deborah Gustafson and Tracey Wilson), U01-HL146202; Data Analysis and Coordination Center (Gypsyamber D'Souza, Stephen Gange and Elizabeth Golub), U01-HL146193; Chicago-Cook County CRS (Mardge Cohen and Audrey French), U01-HL146245; Connie Wofsy Women's HIV Study, Northern California CRS (Bradley Aouizerat and Phyllis Tien), U01-HL146242; Metropolitan Washington CRS (Seble Kassaye and Daniel Merenstein), U01-HL146205; Miami CRS (Maria Alcaide, Margaret Fischl, and Deborah Jones), U01-HL146203; UAB-MS CRS (Mirjam-Colette Kempf and Deborah Konkle-Parker), U01-HL146192; UNC CRS (Adaora Adimora), U01-HL146194. Additionally, the authors would like to thank Jing Wu and Yiwen Wang for their data assembly and analysis efforts.

## Author Contributions

**Conceptualization:** Radoslav Goldman.

**Data curation:** Renhuizi Wei, Brad E. Aouizerat.

**Formal analysis:** Alec T. McIntosh, Renhuizi Wei, Jaeil Ahn.

**Funding acquisition:** Radoslav Goldman.

**Investigation:** Renhuizi Wei, Jaeil Ahn, Brad E. Aouizerat, Seble G. Kassaye, Michael H. Augenbraun, Jennifer C. Price, Audrey L. French, Stephen J. Gange, Kathryn M. Anastos, Radoslav Goldman.

**Methodology:** Alec T. McIntosh, Brad E. Aouizerat, Radoslav Goldman.

**Project administration:** Alec T. McIntosh, Jaeil Ahn, Radoslav Goldman.

**Resources:** Brad E. Aouizerat, Seble G. Kassaye, Michael H. Augenbraun, Jennifer C. Price, Audrey L. French, Stephen J. Gange, Kathryn M. Anastos.

**Software:** Renhuizi Wei.

**Supervision:** Jaeil Ahn, Brad E. Aouizerat, Radoslav Goldman.

**Visualization:** Renhuizi Wei, Jaeil Ahn.

**Writing – original draft:** Alec T. McIntosh.

**Writing – review & editing:** Alec T. McIntosh, Jaeil Ahn, Brad E. Aouizerat, Seble G. Kassaye, Michael H. Augenbraun, Audrey L. French, Stephen J. Gange, Kathryn M. Anastos, Radoslav Goldman.

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
