## [Decision Letter · Decision Letter 0]

8 Dec 2020

PONE-D-20-31305

A genomic variant of *ALPK2* is associated with increased liver fibrosis risk in HIV/HCV coinfected women

PLOS ONE

Dear Dr. Goldman,

Thank you for submitting your manuscript to PLOS ONE. After careful consideration, we feel that it has merit but does not fully meet PLOS ONE’s publication criteria as it currently stands. Therefore, we invite you to submit a revised version of the manuscript that addresses the points raised during the review process.

We look forward to receiving your revised manuscript.

Kind regards,

Jason Blackard, PhD

Academic Editor

PLOS ONE

Journal Requirements:

2. Please provide additional details regarding participant consent.

In the ethics statement in the Methods and online submission information, please ensure that you have specified what type you obtained (for instance, written or verbal, and if verbal, how it was documented and witnessed).

If your study included minors, state whether you obtained consent from parents or guardians.

If the need for consent was waived by the ethics committee, please include this information.

'The MWCCS is funded primarily by the National Heart, Lung, and Blood Institute (NHLBI),

with additional co-funding from the Eunice Kennedy Shriver National Institute Of Child

Health & Human Development (NICHD), National Human Genome Research Institute

(NHGRI), National Institute On Aging (NIA), National Institute Of Dental & Craniofacial Research (NIDCR), National Institute Of Allergy And Infectious Diseases (NIAID), National Institute Of Neurological Disorders And Stroke (NINDS), National Institute Of Mental Health (NIMH), National Institute On Drug Abuse (NIDA), National Institute of Nursing Research (NINR), National Cancer Institute (NCI), National Institute on Alcohol

Abuse and Alcoholism (NIAAA), National Institute on Deafness and Other Communication Disorders (NIDCD), National Institute of Diabetes and Digestive and Kidney Diseases (NIDDK). MWCCS data collection is also supported by UL1-TR000004 (UCSF CTSA), P30-AI-050409 (Atlanta CFAR), P30-AI-050410 (UNC CFAR), and P30-AI-027767 (UAB CFAR).'

'The funders had no role in study design, data collection and analysis, decision to publish, or preparation of the manuscript.'

'The authors of this manuscript have the following competing interests: National Heart, Lung, and Blood Institute (NHLBI; https://www.nhlbi.nih.gov) provided grant support to the following co-authors: U01-HL146242 (BEA, JCP); U01-HL146205 (SGK); U01-HL146202 (MHA); U01-HL146245 (ALF); U01-HL146193 (SJG); and U01-HL146204 (KMA).'

Additional Editor Comments:

This is a study of genetic factors associated with liver fibrosis in HIV/HCV co-infected women.

Data on HIV treatment status, as well as HCV genotype should be included.

In all analyses, HCV mono-infected means HCV RNA positive, correct?

Figure 1 would be easier to view with white/black columns for APRI/FIB-4 rather than the hatched/dashed columns.

Reviewers' comments:

Reviewer's Responses to Questions

**Comments to the Author**

1. Is the manuscript technically sound, and do the data support the conclusions?

Reviewer #1: Partly

Reviewer #2: Yes

2. Has the statistical analysis been performed appropriately and rigorously? 

Reviewer #1: Yes

Reviewer #2: Yes

3. Have the authors made all data underlying the findings in their manuscript fully available?

Reviewer #1: Yes

Reviewer #2: Yes

4. Is the manuscript presented in an intelligible fashion and written in standard English?

Reviewer #1: Yes

Reviewer #2: Yes

5. Review Comments to the Author

Reviewer #1: In this report, Mclntosh AT, et al describes ALPK2 rs3809973 single nucleotide variant

was associated with liver fibrosis in HIV/HCV coinfected patients. These data are very interesting and might be new important knowledge, however there are several concerns as reviewer describes below.

Major points:

1. Authors described in introduction section, HCV and HIV are at high risk of coinfection due to common models of transmission. HBV is also common models of transmission and important and indispensable virus associated with liver fibrosis. You should show present or absence of HBV infection in all your patients.

2. Liver biopsy is certainly an invasive examination, however you should investigate elastography (for example fibroscan or others) to evaluate liver fibrosis.

3. You focused proteins associated with glycosylation. Actually these proteins may associate with liver fibrosis and evaluated in relation to surrogate biomarkers of that. This reviewer cannot understand why you focused only genes associated with glycosylation. You should investigate gene ontology analysis or other and show other gene groups associated with liver fibrosis in your results of nsSNV examination.

4. In Table 1, you should show unit of HIV or HCV load, and also platelet count, ALT, AST needed to calculate FIB-4 or APRI score. Moreover, it is better to show factors associated with hepatic function for example total bilirubin, albumin or prothrombin time.

5. In Table 2, you showed multiple linear regression used SNV, age, PC1-3, HIV load and HCV load. Why do you select these factors? HIV load or HCV load are not associated with degree of liver fibrosis. Moreover, you should show why you select only ALPK2 among SNV listed in this table.

5. You described in method section, you obtained continuous clinical measured of APRI or FIB-4. What time point is data showed in results?

You should also show present or absence of anti-HIV or HCV therapy in your patients.

6. Why do you investigate only female patients? You need to describe differences between female and male patients.

7. Figure 2 and 3 are low resolution and low quality.

Minor point:

In Figure 1, you need to show which group were significantly different.

Reviewer #2: The study is overall well performed and manuscript well written. Authors have suggested a possible mechanism for their positive findings in the discussion section.

However there are major limitations in regards to clinical implications. Some of these have already been identified by authors in the discussion section including one time HCV testing, FIB-4 and APRI not being the gold standard of liver fibrosis, lack of serial testing and high drop out rate, all female cohort.

Other major limitation is that significant confounding factors have not been accounted for. These include duration of HCV infection, presence of other liver disease, in particular, NAFLD which has high prevalence in the general population as well as HIV infected patients, as well chronic Hepatitis B and alcohol consumption. Lack of these data significantly limit the ability to draw definite conclusions regarding association of ALPK2 and liver fibrosis but might be worthwhile exploring in prospective manner.

6. PLOS authors have the option to publish the peer review history of their article (what does this mean?). If published, this will include your full peer review and any attached files.

Reviewer #1: No

Reviewer #2: No

---

## [Author Response · Author response to Decision Letter 0]

18 Jan 2021

Response to Editor:

1. "Data on HIV treatment status, as well as HCV genotype should be included."

We have included HIV treatment status as the Editor has requested. HCV genotype is available for roughly 15% of the coinfected population and none of the HCV monoinfected population. As such, we opted to exclude it from table 1.

2. "In all analyses, HCV mono-infected means HCV RNA positive, correct?" 

Yes, we made sure to note this in the methods section that HCV monoinfection was determined by anti-HCV serum antibodies and detectable HCV RNA titers.

3. "Figure 1 would be easier to view with white/black columns for APRI/FIB-4 rather than the hatched/dashed columns." 

The authors agree and the changes have been made to Figure 1 accordingly.

________

Response to Reviewer #1

1. "Authors described in introduction section, HCV and HIV are at high risk of coinfection due to common models of transmission. HBV is also common models of transmission and important and indispensable virus associated with liver fibrosis. You should show present or absence of HBV infection in all your patients."

The authors are in agreement with the reviewers on the necessity of HBV inclusion. In initial analyses, we excluded it because only 32 patients (2% of the 1,858 participants in the study) were found to have chronic HBV infection. In the revised manuscript, we have incorporated this into our Table 1 and have adjusted for HBV in sensitivity analyses found in Table S3/Figure S2. We observed no major changes in our results and we have added this discussion in Discussion Section. 

2. "Liver biopsy is certainly an invasive examination, however you should investigate elastography (for example fibroscan or others) to evaluate liver fibrosis."

We fully agree that elastography would complement the surrogate fibrosis indices of APRI and FIB-4 presently used. Unfortunately, elastography was not done when patients were recruited. The baseline evaluation predates elastography and is not feasible in this study. Although our surrogate markers of liver fibrosis have their disadvantages, elastography and the associated technology also require some refinement. Several studies have shown that results from elastography can vary from operator to operator on the same individual, and that extremes in patient adiposity can also influence the attainment of a reliable scan/result. As such, the authors find that the surrogate fibrosis indices utilized were practical and appropriate for the time frame and cohort size necessary to investigate the present research questions.

3. "You focused proteins associated with glycosylation. Actually these proteins may associate with liver fibrosis and evaluated in relation to surrogate biomarkers of that. This reviewer cannot understand why you focused only genes associated with glycosylation. You should investigate gene ontology analysis or other and show other gene groups associated with liver fibrosis in your results of nsSNV examination." 

The now combined WIHS/MACS consortium is made up of multiple research groups across the country, as well as multiple sites of data collection. The process for obtaining data for analysis requires the submission and approval of a data request form to the cohort’s executive committee. We were granted the permission to study the glycosylation related genes as described in the manuscript. The authors believe that the selection of SNPs impacting glycosylation and its related processes are highly relevant to infections diseases and understudied at the same time. Finally, as mentioned in the manuscript, not all of the SNPs we requested from the WIHS executive committee were detected in the population, or, were detected but were being investigated by another research group.

4. "In Table 1, you should show unit of HIV or HCV load, and also platelet count, ALT, AST needed to calculate FIB-4 or APRI score. Moreover, it is better to show factors associated with hepatic function for example total bilirubin, albumin or prothrombin time."

The units of HIV and HCV load are denoted in the subtext of Table 1. We attempted to show factors influencing hepatic function in table 1 per the reviewer’s suggestion, however, very few patients had PT/INR, PTT, or total bilirubin reported. We did show average albumin levels, however, this one measure might only be a small viewing window into hepatic function.

5. "In Table 2, you showed multiple linear regression used SNV, age, PC1-3, HIV load and HCV load. Why do you select these factors? HIV load or HCV load are not associated with degree of liver fibrosis. Moreover, you should show why you select only ALPK2 among SNV listed in this table."

The reviewer asks a great question. Factors we adjusted for:

1. SNV: The main factor we were trying to assess for contribution to outcome.

2. Age: Age is built into the FIB-4 liver fibrosis score calculation. Therefore, the authors wanted to ensure that WIHS participants with higher ages did not confound the association of a increased liver fibrosis risk with any given SNV.

3. PC1-3 (Race/Ethnicity): The prevalence of liver fibrosis in various races/ethnicities varies widely. In genome wide association studies, it is becoming commonplace to adjust for race in terms of principal components if available. In our study, PC1-3 explain more than 90% variation

4. HIV load: Increased HIV load, and therefore increased immunosuppression in patients with HIV/HCV coinfection, has been shown previously to be positively correlated with liver fibrosis1,2. Higher HIV viral load equates to more immune dysregulation and a higher chance of liver fibrosis in the coinfected.

5. HCV load: We agree that little evidence is present in the literature that shows HCV RNA load to be associated with increasing severity of liver fibrosis. However, results did not change in SNV identification irrespective of adjusting for chronic HCV infection as a continuous or categorical variable.

1. Pokorska-Śpiewak M, Stańska-Perka A, Popielska J, Ołdakowska A, Coupland U, Zawadka K, Szczepańska-Putz M, Marczyńska M. Prevalence and predictors of liver disease in HIV-infected children and adolescents. Scientific Reports. 2017;7(1):12309. doi:10.1038/s41598-017-11489-2

2. Mohr R, Schierwagen R, Schwarze-Zander C, Boesecke C, Wasmuth J-C, Trebicka J, Rockstroh JK. Liver Fibrosis in HIV Patients Receiving a Modern cART: Which Factors Play a Role? Medicine. 2015;94(50):e2127. doi:10.1097/MD.0000000000002127

What risk factors we have added to the model in the sensitivity analysis:

6. NAFLD: We used Body Mass Index as a continuous surrogate marker in an attempt to adjust for the risk NAFLD poses to increased liver fibrosis risk. Although an imperfect marker for NAFLD (the reason we didn’t incorporate it into the original model) it should help quell the reviewers concerns that we did not adjust for this population of liver disease patients.

7. HBV Infection: We added HBV Status to our list of factors for adjustment. Active HBV infections in the entire sample size equaled to roughly 2%. This low value is ultimately why we didn’t incorporate it into the final model, however it is included in our sensitivity analysis now in the manuscript supplement. 

8. Alcohol Consumption: We used alcohol consumption as a categorical variable and grouped patients into: 1) abstain, 2) mild drinker (1-7drinks per week), and 3) heavy drinker (>7 drinks per week). Self-reported values made the authors a bit nervous, as what is classified as a single alcoholic drink could differ from person to person. Also, people tend to report lower amounts of consumption than reality. This is why alcohol consumption was excluded from the previous model, but has been included in the sensitivity analysis.

What risk factors we didn’t adjust for in the original model or sensitivity analysis:

9. HIV drugs that could cause hepatotoxicity: Since the exact drugs utilized in the anti-viral regimens of participants were not listed, we were unable to determine the impact of the liver toxicity secondary to select HIV drugs within classes. As such, simply accounting for these drugs, stratified into 1) No therapy, 2) Monotherapy, 3) Combination therapy, or 4) HAART, to indirectly account for the possibilities of hepatic toxicity from these drugs without specifics seemed futile. 

Finally, we focused on ALPK2 because it was a variant frequently found in the population and therefore had enough sample size to parse the variant across all serogroups in comparison to the HIV/HCV coinfected. The low frequency of the other SNVs obtained by the linear regression model would not have allowed this to occur. Also, after adjusting for multiple combinations of factors in the linear regression model, the ALPK2 nsSNV was an anchor that did not appear to fluctuate in estimate or significance regardless of the adjustment.

6. "You described in method section, you obtained continuous clinical measured of APRI or FIB-4. What time point is data showed in results? You should also show present or absence of anti-HIV or HCV therapy in your patients."

Our study reports APRI and FIB-4 measures obtained at baseline (or the first visit for each participant), Subsequent measurements were obtained inconsistently, and patient dropout prevented inclusion of these measures at later timepoints. Presence of HIV therapy has been added to Table 1. Data for HCV therapy was not available.

7. "Why do you investigate only female patients? You need to describe differences between female and male patients."

Before the WIHS and male equivalent, Multicenter AIDS Cohort Study (MACS), merged into one cohesive overview committee, data had to be requested from a specific cohort. We selected the WIHS because of the completed genotyping data (absent in MACS) and a general lack of studies of this subject in women. Expanding our findings to HIV/HCV coinfected men will be the grounds for future studies, data requests, and analyses.

8. "Figure 2 and 3 are low resolution and low quality."

Adjustments have been made to improve the resolution and quality of Figures 2 and 3.

9. "In Figure 1, you need to show which group were significantly different."

This action has been completed upon reviewer request. With many comparisons significant in one graph, we elected to only show those that were not significant as opposed to significant. In lieu of this, we have added the more relevant stats upon request.

_________

Response to Reviewer #2

“The study is overall well performed and manuscript well written. Authors have suggested a possible mechanism for their positive findings in the discussion section.

However there are major limitations in regards to clinical implications. Some of these have already been identified by authors in the discussion section including one time HCV testing, FIB-4 and APRI not being the gold standard of liver fibrosis, lack of serial testing and high drop out rate, all female cohort. Other major limitation is that significant confounding factors have not been accounted for. These include duration of HCV infection, presence of other liver disease, in particular, NAFLD which has high prevalence in the general population as well as HIV infected patients, as well chronic Hepatitis B and alcohol consumption. Lack of these data significantly limit the ability to draw definite conclusions regarding association of ALPK2 and liver fibrosis but might be worthwhile exploring in prospective manner.”

The authors appreciate the insight of reviewer 2. Many of the concerns of this reviewer have been newly incorporated into our model (HBV, BMI (Surrogate for NAFLD), and Alcohol consumption) in a sensitivity analysis. We observe that with our original list of SNV targets being upheld after these additional adjustments (Please refer to Supplementary Table 3/Table 2). For those factors not addressed, reviewer 2 offers many ways that have allowed the authors to expand their discussion section regarding the limitations of the study.

---

## [Decision Letter · Decision Letter 1]

4 Feb 2021

A genomic variant of *ALPK2* is associated with increased liver fibrosis risk in HIV/HCV coinfected women

PONE-D-20-31305R1

Dear Dr. Goldman,

We’re pleased to inform you that your manuscript has been judged scientifically suitable for publication and will be formally accepted for publication once it meets all outstanding technical requirements.

Kind regards,

Jason T. Blackard, PhD

Academic Editor

PLOS ONE

Additional Editor Comments (optional):

None

Reviewers' comments:

Reviewer's Responses to Questions

**Comments to the Author**

1. If the authors have adequately addressed your comments raised in a previous round of review and you feel that this manuscript is now acceptable for publication, you may indicate that here to bypass the “Comments to the Author” section, enter your conflict of interest statement in the “Confidential to Editor” section, and submit your "Accept" recommendation.

Reviewer #1: All comments have been addressed

Reviewer #2: All comments have been addressed

2. Is the manuscript technically sound, and do the data support the conclusions?

Reviewer #1: Yes

Reviewer #2: Yes

3. Has the statistical analysis been performed appropriately and rigorously? 

Reviewer #1: Yes

Reviewer #2: Yes

4. Have the authors made all data underlying the findings in their manuscript fully available?

Reviewer #1: Yes

Reviewer #2: Yes

5. Is the manuscript presented in an intelligible fashion and written in standard English?

Reviewer #1: Yes

Reviewer #2: Yes

6. Review Comments to the Author

Reviewer #1: (No Response)

Reviewer #2: Despite some limitations in the data due to the nature of the study, authors have addressed most of the major concerns

7. PLOS authors have the option to publish the peer review history of their article (what does this mean?). If published, this will include your full peer review and any attached files.

Reviewer #1: No

Reviewer #2: No

---

## [Editor Report · Acceptance letter]

2 Mar 2021

PONE-D-20-31305R1 

A genomic variant of *ALPK2* is associated with increased liver fibrosis risk in HIV/HCV coinfected women 

Dear Dr. Goldman:

I'm pleased to inform you that your manuscript has been deemed suitable for publication in PLOS ONE. Congratulations! Your manuscript is now with our production department. 

Kind regards, 

on behalf of

Dr. Jason T. Blackard 

Academic Editor

PLOS ONE